# CANE: Context-Aware Network Embedding for Relation Modeling

## Abstract

Network embedding (NE) is playing a critical role in network analysis, due to its ability to represent vertices with efficient low-dimensional embedding vectors. However, existing NE models aim to learn a fixed context-free embedding for each vertex, and neglect the diverse roles when interacting with other vertices. In this paper, we assume that one vertex usually shows different aspects when interacting with different neighbor vertices, and should own different embeddings respectively. Therefore, we present Context-Aware Network Embedding (CANE), a novel NE model to address this issue. CANE learns context-aware embeddings for vertices with mutual attention mechanism and is expected to model the semantic relationships between vertices more precisely. In experiments, we compare our model with existing NE models on three real-world datasets. Experimental results shows that CANE achieves significant improvement than state-of-the-art methods on link prediction, and comparable performance on vertex classification.

## 1 Introduction

Network embedding (NE), i.e., network representation learning (NRL), aims to map vertices of a network into a low-dimensional space according to their structural roles in the network. NE provides an efficient and effective way to represent and manage large-scale networks, alleviating the computation and sparsity issues of conventional symbol-based representations. Hence, NE is attracting many research interests in recent years (Perozzi et al., 2014; Tang et al., 2015; Grover and Leskovec, 2016), and achieves promising performance on many network analysis tasks including link prediction, vertex classification, and community detection.

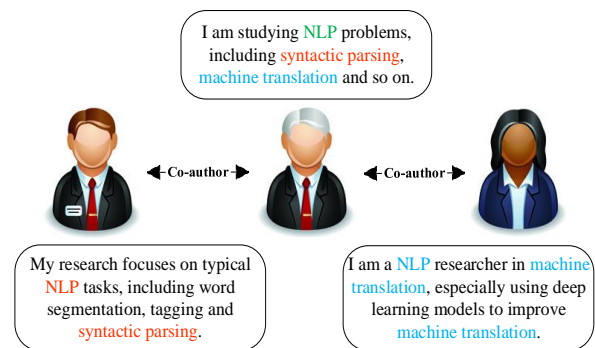

Figure 1: Example of text-based information network. (Red, blue and green fonts represent concerns of the left user, right user and both respectively.)

In real-world social networks, it is intuitive that one vertex may demonstrate various aspects when interacting with different neighbor vertices. For example, a researcher usually collaborates with different partners on diverse research topics (as illustrated in Fig. 1), a social-media user contacts with various friends sharing distinct interests, and a web page links with multiple pages for different purposes. However, most existing NE methods only arrange one single embedding vector to each vertex, and give rise to the following two invertible issues: (1) These methods cannot cope with the aspect transition of a vertex flexibly when interacting with different neighbors. (2) In these models, a vertex tends to force the embeddings of its neighbors close to each other, which may be not case all the time. For example, the left user and right user in Fig. 1, share less common interests, but are learned to be close to each other since they both link to the middle person. This will accordingly

make vertex embeddings indiscriminative.

To address these issues, we aim to propose a **C**ontext-**A**ware **N**etwork **E**mbedding (CANE) framework for modeling relationships between vertices precisely. More specifically, we present CANE on information networks, where each vertex also contains rich external information such as text, labels or other meta-data, and the significance of context is more critical for NE in this scenario. Without loss of generality, we implement CANE on text-based information networks in this paper, which can be easily extended to other types of information networks.

In conventional NE models, each vertex is represented as a static embedding vector, denoted as **context-free embedding**. On the contrary, CANE assigns dynamic embeddings to a vertex according to different neighbors it interacts with, named as **context-aware embedding** . Take a vertex $u$ and its neighbor vertex $v$ for example. The context-free embedding of $u$ remains unchanged when interacting with different neighbors. On the contrary, the context-aware embedding of $u$ is dynamic when confronting different neighbors.

When $u$ interacting with $v$, their context embeddings with respect to each other are derived from their text information, $S_u$ and $S_v$ respectively. For each vertex, we can easily use neural models, such as convolutional neural networks (Blunsom et al., 2014; Johnson and Zhang, 2014; Kim, 2014) and recurrent neural networks (Kiros et al., 2015; Tai et al., 2015), to build **context-free** text-based embedding. In order to realize **context-aware** text-based embeddings, we introduce the selective attention scheme and build **mutual attention** between $u$ and $v$ into these neural models. The mutual attention is expected to guide neural models to emphasize those words that are focused by its neighbor vertices and eventually obtain context-aware embeddings.

Both context-free embeddings and context-aware embeddings of each vertex can be efficiently learned together via concatenation using existing NE methods such as DeepWalk (Perozzi et al., 2014), LINE (Tang et al., 2015) and node2vec (Grover and Leskovec, 2016).

We conduct experiments on three real-world datasets of different areas. Experimental results on link prediction reveal the effectiveness of our framework as compared to other state-of-the-art methods. The results suggest that, context-aware embeddings are critical for network analysis, especially for those tasks concerning about complicated interactions between vertices such as link prediction. We also explore the performance of our framework via vertex classification and case studies, which again confirms the flexibility and superiority of our models.

## 2 Related Work

With the rapid growth of large-scale social networks, network embedding, i.e. network representation learning has been proposed as a critical technique for network analysis tasks.

In recent years, there have been a large number of NE models proposed to learn efficient vertex embeddings (Tang and Liu, 2009; Cao et al., 2015; Wang et al., 2016). For example, DeepWalk (Perozzi et al., 2014) performs random walks over networks and introduces an efficient word representation learning model, Skip-Gram (Mikolov et al., 2013a), to learn network embeddings. LINE (Tang et al., 2015) optimizes the joint and conditional probabilities of edges in large-scale networks to learn vertex representations. Node2vec (Grover and Leskovec, 2016) modifies the random walk strategy in DeepWalk into biased random walks to explore the network structure more efficiently. Nevertheless, most of these NE models only encode the structural information into vertex embeddings, without considering heterogenous information accompanied with vertices in real-world social networks.

To address this issue, researchers make great efforts to incorporate heterogenous information into conventional NE models. For instance, Yang et al. (2015) present text-associated DeepWalk (TADW) to improve matrix factorization based DeepWalk with text information. Tu et al. (2016) propose max-margin DeepWalk (MMDW) to learn discriminative network representations by utilizing labeling information of vertices. Chen et al. (2016) propose group-enhanced network embedding (GENE) to integrate existing group information in NE. Sun et al. (2016) regard text content as a special kind of vertices, and propose context-enhanced network embedding (CENE) through leveraging both structural and textural information to learn network embeddings.

To the best of our knowledge, all existing NE models focus on learning context-free embeddings, but ignore the diverse roles when a vertex

interacts with others. In contrast, we assume that a vertex has different embeddings according to which vertex it interacts with, and propose CANE to learn context-aware vertex embeddings.

## 3 Problem Formulation

We first give basic notations and definitions in this work. Suppose there is an information network $G = (V, E, T)$, where $V$ is the set of vertices, $E \subseteq V \times V$ are edges between vertices, and $T$ denotes the text information of vertices. Each edge $e_{u,v} \in E$ represents the relationship between two vertices $(u, v)$, with an associated weight $w_{u,v}$. Here, the text information of a specific vertex $v \in V$ is represented as a word sequence $S_v = (w_1, w_2, \ldots, w_{n_v})$, where $n_v = |S_v|$. NRL aims to learn a low-dimensional embedding $\mathbf{v} \in \mathbb{R}^d$ for each vertex $v \in V$ according to its network structure and associated information, e.g. text and labels. Note that, $d \ll |V|$ is the dimension of representation space.

**Definition 1. Context-free Embeddings:** Conventional NRL models learn context-free embedding for each vertex. It means the embedding of a vertex is fixed and won't change with respect to its context information (i.e., another vertex it interacts with).

**Definition 2. Context-aware Embeddings:** Different from existing NRL models that learn context-free embeddings, CANE learns various embeddings for a vertex according to its different contexts. Specifically, for an edge $e_{u,v}$, CANE learns context-aware embeddings $\mathbf{v}_{(u)}$ and $\mathbf{u}_{(v)}$.

## 4 The Method

### 4.1 Overall Framework

To take full use of both network structure and associated text information, we propose two types of embeddings for a vertex $v$, i.e., structure-based embedding $\mathbf{v}^s$ and text-based embedding $\mathbf{v}^t$. Structure-based embedding is able to capture the information in the network structure, while text-based embedding can capture the textual meanings lying in the associated text information. With these embeddings, we can simply concatenate them and obtain the vertex embeddings as $\mathbf{v} = \mathbf{v}^s \oplus \mathbf{v}^t$, where $\oplus$ indicates the concatenation operation. Note that, the text-based embedding $\mathbf{v}^t$ can be either context-free or context-aware, which will be introduced detailedly in section 4.4 and 4.5

respectively. When $\mathbf{v}^t$ is context-aware, the overall vertex embeddings $\mathbf{v}$ will be context-aware as well.

With above definitions, CANE aims to maximize the overall objective of edges as follows:

$$\mathcal{L} = \sum_{e \in E} L(e). \qquad (1)$$

Here, the objective of each edge $L(e)$ consists of two parts as follows:

$$L(e) = L_s(e) + L_t(e), \qquad (2)$$

where $L_s(e)$ denotes the structure-based objective and $L_t(e)$ represents the text-based objective.

In the following part, we give detailed introduction to the two objectives respectively.

### 4.2 Structure-based Objective

Without loss of generality, we assume the network is directed, as an undirected edge can be considered as two directed edges with opposite directions and equal weights.

Thus, the structure-based objective aims to measure the log-likelihood of a directed edge using the structure-based embeddings as

$$L_s(e) = w_{u,v} \log p(\mathbf{v}^s | \mathbf{u}^s). \qquad (3)$$

Following LINE (Tang et al., 2015), we define the conditional probability of $v$ generated by $u$ in Eq. (3) as

$$p(\mathbf{v}^s | \mathbf{u}^s) = \frac{\exp(\mathbf{u}^s \cdot \mathbf{v}^s)}{\sum_{z \in V} \exp(\mathbf{u}^s \cdot \mathbf{z}^s)}. \qquad (4)$$

### 4.3 Text-based Objective

Vertices in real-world social networks usually accompany with associated text information. Therefore, we propose the text-based objective to take advantage of these text information, as well as learn text-based embeddings for vertices.

The text-based objective $L_t(e)$ can be defined with various measurements. To be compatible with $L_s(e)$, we define $L_t(e)$ as follows:

$$L_t(e) = \alpha \cdot L_{tt}(e) + \beta \cdot L_{ts}(e) + \gamma \cdot L_{st}(e), \qquad (5)$$

where $\alpha$, $\beta$ and $\gamma$ control the weights of various parts, and

$$\begin{aligned} L_{tt}(e) &= w_{u,v} \log p(\mathbf{v}^t | \mathbf{u}^t), \\ L_{ts}(e) &= w_{u,v} \log p(\mathbf{v}^t | \mathbf{u}^s), \qquad (6) \\ L_{st}(e) &= w_{u,v} \log p(\mathbf{v}^s | \mathbf{u}^t). \end{aligned}$$

The conditional probabilities in Eq. (6) map the two types of vertex embeddings into the same representation space, but do not enforce them to be identical for the consideration of their own characteristics. Similarly, we employ softmax function for calculating the probabilities, as in Eq. (4).

The structure-based embeddings are regarded as parameters, the same as in conventional NE models. But for text-based embeddings, we intend to obtain them from associated text information of vertices. Besides, the text-based embeddings can be obtained either in context-free ways or in context-aware ones. In the following sections, we will give detailed introduction respectively.

### 4.4 Context-Free Text Embedding

There has been a variety of neural models to obtain text embeddings from a word sequence, such as convolutional neural networks (CNN) (Blunsom et al., 2014; Johnson and Zhang, 2014; Kim, 2014) and recurrent neural networks (RNN) (Kiros et al., 2015; Tai et al., 2015).

In this work, we investigate different neural networks for text modeling, including CNN, Bidirectional RNN (Schuster and Paliwal, 1997) and GRU (Cho et al., 2014), and employ the best performed CNN, which can capture the local semantic dependency among words.

Taking the word sequence of a vertex as input, CNN obtains the text-based embedding through three layers, i.e. looking-up, convolution and pooling.

**Looking-up.** Given a word sequence $S = (w_1, w_2, \ldots, w_n)$, the looking-up layer transforms each word $w_i \in S$ into its corresponding word embedding $\mathbf{w}_i \in \mathbb{R}^{d'}$ and obtains embedding sequence as $\mathbf{S} = (\mathbf{w}_1, \mathbf{w}_2, \ldots, \mathbf{w}_n)$. Here, $d'$ indicates the dimension of word embeddings.

**Convolution.** After looking-up, the convolution layer extracts local features of input embedding sequence $\mathbf{S}$. To be specific, it performs convolution operation over a sliding window of length $l$ using a convolution matrix $\mathbf{C} \in \mathbb{R}^{d \times (l \times d')}$ as follows:

$$\mathbf{x}_i = \mathbf{C} \cdot \mathbf{S}_{i:i+l-1} + \mathbf{b}, \qquad (7)$$

where $\mathbf{S}_{i:i+l-1}$ denotes the concatenation of word embeddings within the $i$-th window and $\mathbf{b}$ is the bias vector. Note that, we add zero padding vectors (Hu et al., 2014) at the edge of the sentence.

**Max-pooling.** To obtain the text embedding $\mathbf{v}^t$, we operate max-pooling and non-linear transfor-

mation over $\{\mathbf{x}_0^i, \ldots, \mathbf{x}_n^i\}$ as follows:

$$r_i = \tanh(\max(\mathbf{x}_0^i, \ldots, \mathbf{x}_n^i)), \qquad (8)$$

At last, we encode the text information of a vertex with CNN and obtain its text-based embedding $\mathbf{v}^t = [r_1, \ldots, r_d]^T$. As $\mathbf{v}^t$ is irrelevant to the other vertices it interacts with, we name it as context-free text embedding.

### 4.5 Context-Aware Text Embedding

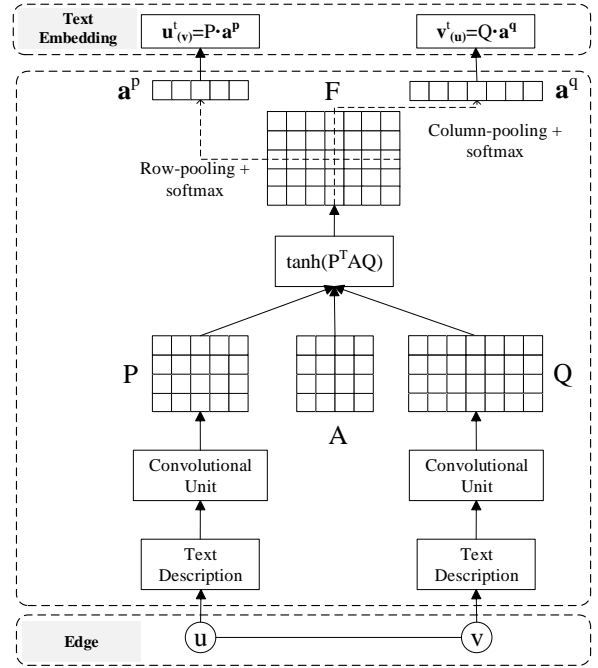

Figure 2: An illustration of context-aware text embedding.

As stated before, we assume that a specific vertex plays different roles when interacting with others vertices. In other words, each vertex should have its own points of focus about a specific vertex, which leads to its context-aware text embeddings.

To achieve this, we employ **mutual attention** to obtain context-aware text embedding. It enables the pooling layer in CNN to be aware of the vertex pair in an edge, in a way that text information from a vertex can directly affect the text embedding of the other vertex, and vice versa.

In Fig. 2, we give an illustration of the generating process of context-aware text embedding. Given an edge $e_{u,v}$ with two corresponding text sequences $S_u$ and $S_v$, we can get the matrices $\mathbf{P} \in \mathbb{R}^{d \times m}$ and $\mathbf{Q} \in \mathbb{R}^{d \times n}$ through convolution layer. Here, $m$ and $n$ represent the lengths of $S_u$

and $S_v$ respectively. By introducing an attentive matrix $\mathbf{A} \in \mathbb{R}^{d \times d}$, we compute the correlation matrix $\mathbf{F} \in \mathbb{R}^{m \times n}$ as follows:

$$\mathbf{F} = \tanh(\mathbf{P}^T \mathbf{A} \mathbf{Q}). \quad (9)$$

Note that, each element $\mathbf{F}_{i,j}$ in $\mathbf{F}$ represents the pair-wise correlation score between two hidden vectors, i.e., $\mathbf{P}_i$ and $\mathbf{Q}_j$.

After that, we conduct pooling operations along rows and columns of $\mathbf{F}$ to generate the importance vectors, named as row-pooling and column pooling respectively. According to our experiments, mean-pooling performs better than max-pooling. Thus, we employ mean-pooling operation as follows:

$$\begin{aligned} g_i^p &= \mathbf{mean}(\mathbf{F}_{i,1}, \dots, \mathbf{F}_{i,n}), \\ g_i^q &= \mathbf{mean}(\mathbf{F}_{1,i}, \dots, \mathbf{F}_{m,i}). \end{aligned} \quad (10)$$

The importance vectors of $\mathbf{P}$ and $\mathbf{Q}$ are obtained as $\mathbf{g}^p = [g_1^p, \dots, g_m^p]^T$ and $\mathbf{g}^q = [g_1^q, \dots, g_n^q]^T$.

Next, we employ softmax function to transform importance vectors $\mathbf{g}^p$ and $\mathbf{g}^q$ to attention vectors $\mathbf{a}^p$ and $\mathbf{a}^q$. For instance, the $i$-th element of $\mathbf{a}^p$ is formalized as follows:

$$a_i^p = \frac{\exp(g_i^p)}{\sum_{j \in [1,m]} \exp(g_j^p)}. \quad (11)$$

At last, the context-aware text embeddings of $u$ and $v$ are computed as

$$\begin{aligned} \mathbf{u}_{(v)}^t &= \mathbf{P}\mathbf{a}^p, \\ \mathbf{v}_{(u)}^t &= \mathbf{Q}\mathbf{a}^q. \end{aligned} \quad (12)$$

Now, given an edge $(u, v)$, we can obtain the context-aware embeddings of vertices with their structure embeddings and context-aware text embeddings as $\mathbf{u}_{(v)} = \mathbf{u}^s \oplus \mathbf{u}_{(v)}^t$ and $\mathbf{v}_{(u)} = \mathbf{v}^s \oplus \mathbf{v}_{(u)}^t$.

## 4.6 Optimization of CANE

According to Eq. (3) and Eq. (6), CANE aims to maximize several conditional probabilities between $\mathbf{u} \in \{\mathbf{u}^s, \mathbf{u}_{(v)}^t\}$ and $\mathbf{v} \in \{\mathbf{v}^s, \mathbf{v}_{(u)}^t\}$. It is intuitive that optimizing the conditional probability using softmax function is computationally expensive. Thus, we employ negative sampling (Mikolov et al., 2013b) and transform the objective into the following form:

$$\log \sigma(\mathbf{u}^T \cdot \mathbf{v}) + \sum_{i=1}^{k} E_{z \sim P(v)}[\log \sigma(-\mathbf{u}^T \cdot \mathbf{z})], \quad (13)$$

where $k$ is the number of negative samples and $\sigma$ represents the sigmoid function. $P(v) \propto d_v^{3/4}$ denotes the distribution of vertices, where $d_v$ is the out-degree of $v$.

Afterwards, we employ Adam (Kingma and Ba, 2015) to optimize the transformed objective.

## 5 Experiments

In order to investigate the effectiveness of CANE on modeling relationships between vertices, we conduct experiments of link prediction on several real-world datasets. Besides, we also employ vertex classification to verify whether context-aware embeddings of a vertex can compose a high-quality context-free embedding in return.

### 5.1 Datasets

| Datasets | Cora | HepTh | Zhihu |
|---|---|---|---|
| #Vertices | 2,277 | 1,038 | 10,000 |
| #Edges | 5,214 | 1,990 | 43,894 |
| #Labels | 7 | − | − |

Table 1: Statistics of Datasets.

We select three real-world network datasets as follows:

**Cora**[1] is a typical paper citation network constructed by (McCallum et al., 2000). After filtering out papers without text information, there are 2,277 machine learning papers in this network, which are divided into 7 categories.

**HepTh**[2] (High Energy Physics Theory) is another citation network from arXiv[3] released by (Leskovec et al., 2005). We filter out papers without abstract information and retain 1,038 papers at last.

**Zhihu**[4] is the largest online Q&A website in China. Users follow each other and answer questions in this site. We randomly crawl 10,000 active users from Zhihu, and take the descriptions of their concerned topics as text information.

The detailed statistics are listed in Table 1.

### 5.2 Baselines

We employ the following methods as baselines:

**Structure-only:**

*DeepWalk* (Perozzi et al., 2014) performs random walks over networks and employ Skip-Gram

---

[1]https://people.cs.umass.edu/~mccallum/data.html
[2]https://snap.stanford.edu/data/cit-HepTh.html
[3]https://arxiv.org/
[4]https://www.zhihu.com/

| %Removed edges | 15% | 25% | 35% | 45% | 55% | 65% | 75% | 85% | 95% |
|---|---|---|---|---|---|---|---|---|---|
| DeepWalk | 56.0 | 63.0 | 70.2 | 75.5 | 80.1 | 85.2 | 85.3 | 87.8 | 90.3 |
| LINE | 55.0 | 58.6 | 66.4 | 73.0 | 77.6 | 82.8 | 85.6 | 88.4 | 89.3 |
| node2vec | 55.9 | 62.4 | 66.1 | 75.0 | 78.7 | 81.6 | 85.9 | 87.3 | 88.2 |
| Naive Combination | 72.7 | 82.0 | 84.9 | 87.0 | 88.7 | 91.9 | 92.4 | 93.9 | 94.0 |
| TADW | 86.6 | 88.2 | 90.2 | 90.8 | 90.0 | 93.0 | 91.0 | 93.4 | 92.7 |
| CENE | 72.1 | 86.5 | 84.6 | 88.1 | 89.4 | 89.2 | 93.9 | 95.0 | 95.9 |
| CANE (text only) | 78.0 | 80.5 | 83.9 | 86.3 | 89.3 | 91.4 | 91.8 | 91.4 | 93.3 |
| CANE (w/o attention) | 85.8 | 90.5 | 91.6 | 93.2 | 93.9 | 94.6 | 95.4 | 95.1 | 95.5 |
| CANE | **86.8** | **91.5** | **92.2** | **93.9** | **94.6** | **94.9** | **95.6** | **96.6** | **97.7** |

Table 2: AUC values on Cora. ($\alpha = 1.0, \beta = 0.3, \gamma = 0.3$)

model (Mikolov et al., 2013a) to learn vertex embeddings.

*LINE* (Tang et al., 2015) learns vertex embeddings in large-scale networks using first-order and second-order proximities.

*Node2vec* (Grover and Leskovec, 2016) proposes a biased random walk algorithm based on DeepWalk to explore neighborhood architecture more efficiently.

**Structure and Text:**

*Naive Combination:* We simply concatenate the best-performed structure-based embeddings with CNN based embeddings to represent the vertices.

*TADW* (Yang et al., 2015) employs matrix factorization to incorporate text features of vertices into network embeddings.

*CENE* (Sun et al., 2016) leverages both structure and textural information by regarding text content as a special kind of vertices, and optimizes the probabilities of heterogenous links.

### 5.3 Evaluation Metrics and Experiment Settings

For link prediction, we adopt a standard evaluation metric **AUC** (Hanley and McNeil, 1982), which represents the probability that vertices in a random unobserved link are more similar than those in a random nonexistent link.

For vertex classification, we employ L2-regularized logistic regression (L2R-LR) (Fan et al., 2008) to train classifiers, and evaluate the classification accuracies of various methods.

To be fair, we set the embedding dimension to 200 for all methods. In LINE, we set the number of negative samples to 5; we learn the 100 dimensional first-order and second-order embeddings respectively, and concatenate them to form the 200 dimensional embeddings. In node2vec, we employ grid search and select the best per-

formed hyper-parameters for training. We also employ grid search to set the hyper-parameters $\alpha$, $\beta$ and $\gamma$ in CANE. Besides, we set the number of negative samples $k$ to 1 in CANE to speed up the training process. To demonstrate the effectiveness of considering attention mechanism and two types of objectives in Eqs. (3) and (6), we design three versions of CANE for evaluation, i.e., CANE with text only, CANE without attention and CANE.

### 5.4 Link Prediction

As shown in Table 2, Table 3 and Table 4, we evaluate the AUC values while removing different ratios of edges on Cora, HepTh and Zhihu respectively. Note that, when we only keep 5% edges for training, most vertices are isolated, which results in the poor and meaningless performance of all the methods. Thus, we omit the results under this training ratio. From these tables, we have the following observations:

(1) Our proposed CANE consistently achieves significant improvement comparing to all the baselines on all different datasets and different training ratios. It indicates the effectiveness of CANE when applied to link prediction task, and verifies that CANE has the capability of modeling relationships between vertices precisely.

(2) What calls for special attention is that, both CENE and TADW exhibit unstable performance under various training ratios. Specifically, CENE performs poorly under small training ratios, because it reserves much more parameters (e.g., convolution kernels and word embeddings) than TADW, which need more data for training. Different from CENE, TADW performs much better under small training ratios, because DeepWalk based methods can explore the sparse network structure well through random walks even with limited edges. However, it achieves poor performance

| %Removed edges | 15% | 25% | 35% | 45% | 55% | 65% | 75% | 85% | 95% |
|---|---|---|---|---|---|---|---|---|---|
| DeepWalk | 55.2 | 66.0 | 70.0 | 75.7 | 81.3 | 83.3 | 87.6 | 88.9 | 88.0 |
| LINE | 53.7 | 60.4 | 66.5 | 73.9 | 78.5 | 83.8 | 87.5 | 87.7 | 87.6 |
| node2vec | 57.1 | 63.6 | 69.9 | 76.2 | 84.3 | 87.3 | 88.4 | 89.2 | 89.2 |
| Naive Combination | 78.7 | 82.1 | 84.7 | 88.7 | 88.7 | 91.8 | 92.1 | 92.0 | 92.7 |
| TADW | 87.0 | 89.5 | 91.8 | 90.8 | 91.1 | 92.6 | 93.5 | 91.9 | 91.7 |
| CENE | 86.2 | 84.6 | 89.8 | 91.2 | 92.3 | 91.8 | 93.2 | 92.9 | 93.2 |
| CANE (text only) | 83.8 | 85.2 | 87.3 | 88.9 | 91.1 | 91.2 | 91.8 | 93.1 | 93.5 |
| CANE (w/o attention) | 84.5 | 89.3 | 89.2 | 91.6 | 91.1 | 91.8 | 92.3 | 92.5 | 93.6 |
| CANE | **90.0** | **91.2** | **92.0** | **93.0** | **94.2** | **94.6** | **95.4** | **95.7** | **96.3** |

Table 3: AUC values on HepTh. ($\alpha = 0.7, \beta = 0.2, \gamma = 0.2$)

| %Removed edges | 15% | 25% | 35% | 45% | 55% | 65% | 75% | 85% | 95% |
|---|---|---|---|---|---|---|---|---|---|
| DeepWalk | 56.6 | 58.1 | 60.1 | 60.0 | 61.8 | 61.9 | 63.3 | 63.7 | 67.8 |
| LINE | 52.3 | 55.9 | 59.9 | 60.9 | 64.3 | 66.0 | 67.7 | 69.3 | 71.1 |
| node2vec | 54.2 | 57.1 | 57.3 | 58.3 | 58.7 | 62.5 | 66.2 | 67.6 | 68.5 |
| Naive Combination | 55.1 | 56.7 | 58.9 | 62.6 | 64.4 | 68.7 | 68.9 | 69.0 | 71.5 |
| TADW | 52.3 | 54.2 | 55.6 | 57.3 | 60.8 | 62.4 | 65.2 | 63.8 | 69.0 |
| CENE | 56.2 | 57.4 | 60.3 | 63.0 | 66.3 | 66.0 | 70.2 | 69.8 | 73.8 |
| CANE (text only) | 55.6 | 56.9 | 57.3 | 61.6 | 63.6 | 67.0 | 68.5 | 70.4 | 73.5 |
| CANE (w/o attention) | 56.7 | 59.1 | 60.9 | 64.0 | 66.1 | 68.9 | 69.8 | 71.0 | 74.3 |
| CANE | **56.8** | **59.3** | **62.9** | **64.5** | **68.9** | **70.4** | **71.4** | **73.6** | **75.4** |

Table 4: AUC values on Zhihu. ($\alpha = 1.0, \beta = 0.3, \gamma = 0.3$)

under large ones, as its simplicity and the limitation of bag-of-words assumption. On the contrary, CANE has a stable performance on various situations. It demonstrates the flexibility and robustness of CANE.

(3) By introducing attention mechanism, the learnt context-aware embeddings obtain considerable improvements than the ones without attention. It verifies our assumption that a specific vertex should play different roles when interacting with other vertices, and thus benefits the relevant link prediction task.

To summarize, all the above observations demonstrate that CANE is able to learn high-quality context-aware embeddings, which are conducive to estimating the relationship between vertices precisely. Moreover, the experimental results on link prediction task state the effectiveness and robustness of CANE.

## 5.5 Vertex Classification

In CANE, we obtain various embeddings of a vertex according to the vertex it connects to. It's intuitive that the obtained context-aware embeddings are naturally applicable to link prediction task. However, network analysis tasks, such as vertex classification and clustering, require a global embedding, rather than several context-aware embeddings for each vertex.

To demonstrate the capability of CANE to solve these issues, we generate the global embedding of a vertex $u$ by simply averaging all the context-aware embeddings as follows:

$$\mathbf{u} = \frac{1}{N} \sum_{(u,v)|(v,u)\in E} \mathbf{u}_{(v)},$$

where $N$ indicates the number of context-aware embeddings of $u$.

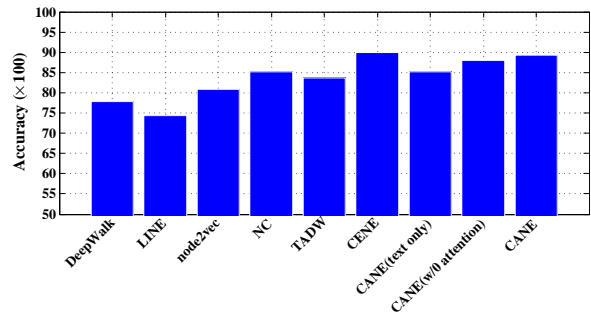

Figure 3: Vertex classification results on Cora.

With the generated global embeddings, we conduct experiments of vertex classification on Cora. As shown in Fig. 3, we observe that:

(1) CANE achieves comparable performance with state-of-the-art model CENE. It states that the

learnt context-aware embeddings can transform into high-quality context-free embeddings through simple average operation, which can be further employed to other network analysis tasks.

(2) With the introduction of mutual attention mechanism, CANE has an encouraging improvement than the one without attention, which is in accordance with the results of link prediction. It denotes that CANE is flexible to various network analysis tasks.

## 5.6 Case Study

To demonstrate the significance of **mutual attention** on selecting meaningful features from text information, we visualize the heat maps of two vertex pairs in Fig. 4. Note that, every word in this figure accompanies with various background colors. The stronger the background color is, the larger the weight of this word is. The weight of each word is calculated according to the attention weights as follows.

For each vertex pair, we can get the attention weight of each convolution window according to Eq. (11). To obtain the weights of words, we assign the attention weight to each word in this window, and add the attention weights of a word together as its final weight.

We select three connected vertices in Cora for example, denoted as A, B and C. From Fig. 4, we observe that, though there exists citation relations with identical paper A, paper B and C concern about different parts of A. The attention weights over A in edge #1 are assigned to "reinforcement learning". On the contrary, the weights in edge #2 are assigned to "machine learning'", "supervised learning algorithms" and "complex stochastic models". Moreover, all these key elements in A can find corresponding words in B and C. It's intuitive that these key elements give an exact explanation on the citation relations. The discovered significant correlations between vertex pairs reflects the effectiveness of mutual attention mechanism, as well as the capability of CANE for modeling relations precisely.

## 6 Conclusion and Future Work

In this paper, we propose the concept of Context-Aware Network Embedding (CANE) for the first time, which aims to learn various context-aware embeddings for a vertex according to the neighbors it interacts with. Specifically, we implement

**Edge #1: (A, B)**
Machine Learning research making great progress many directions This article summarizes four directions discusses current open problems The four directions improving classification accuracy learning ensembles classifiers methods scaling supervised learning algorithms reinforcement learning learning complex stochastic models

The problem making optimal decisions uncertain conditions central Artificial Intelligence If state world known times world modeled Markov Decision Process MDP MDPs studied extensively many methods known determining optimal courses action policies The realistic case state information partially observable Partially Observable Markov Decision Processes POMDPs received much less attention The best exact algorithms problems inefficient space time We introduce Smooth Partially Observable Value Approximation SPOVA new approximation method quickly yield good approximations improve time This method combined reinforcement learning methods combination effective test cases

**Edge #2: (A, C)**
Machine Learning research making great progress many directions This article summarizes four directions discusses current open problems The four directions improving classification accuracy learning ensembles classifiers methods scaling supervised learning algorithms reinforcement learning learning complex stochastic models

In context machine learning examples paper deals problem estimating quality attributes without dependencies among Kira Rendell developed algorithm called RELIEF shown efficient estimating attributes Original RELIEF deal discrete continuous attributes limited twoclass problems In paper RELIEF analysed extended deal noisy incomplete multiclass data sets The extensions verified various artificial one well known realworld problem

Figure 4: Visualizations of mutual attention.

CANE on text-based information networks with proposed mutual attention mechanism, and conduct experiments on several real-world information networks. Experimental results on link prediction demonstrate that CANE is effective for modeling the relationship between vertices. Besides, the learnt context-aware embeddings can compose high-quality context-free embeddings.

We will explore the following directions in future:

(1) We have investigated the effectiveness of CANE on text-based information networks. In future, we will strive to implement CANE on wider variety of information networks with multi-modal data, such as labels, images and so on.

(2) CANE encodes latent relations between vertices into their context-aware embeddings. Furthermore, there usually exist explicit relations in social networks (e.g., families, friends and colleagues relations between social network users), which are expected to be critical to NE. Thus, we want to explore how to incorporate and predict these explicit relations between vertices in NE.

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
