# Peer review of "CANE: Context-Aware Network Embedding for Relation Modeling"

_ACL 2017 — decision unknown_

[Official Review · Reviewer 1 · rating 4 · confidence 4]
soundness 5 · originality 5 · clarity 4 · impact 3 · substance 4 · appropriateness 5 · meaningful comparison 3 · presentation format Poster

This paper addresses the network embedding problem by introducing a neural
network model which uses both the network structure and associated text on the
nodes, with an attention model to vary the textual representation based on the
text of the neighboring nodes.

- Strengths:

The model leverages both the network and the text to construct the latent
representations, and the mutual attention approach seems sensible.

A relatively thorough evaluation is provided, with multiple datasets,
baselines, and evaluation tasks.

- Weaknesses:

Like many other papers in the "network embedding" literature, which use neural
network techniques inspired by word embeddings to construct latent
representations of nodes in a network, the previous line of work on
statistical/probabilistic modeling of networks is ignored.  In particular, all
"network embedding" papers need to start citing, and comparing to, the work on
the latent space model of Peter Hoff et al., and subsequent papers in both
statistical and probabilistic machine learning publication venues:

P.D. Hoff, A.E. Raftery, and M.S. Handcock. Latent space approaches to social
network analysis. J. Amer. Statist. Assoc., 97(460):1090–1098, 2002.

This latent space network model, which embeds each node into a low-dimensional
latent space, was written as far back as 2002, and so it far pre-dates neural
network-based network embeddings.

Given that the aim of this paper is to model differing representations of
social network actors' different roles, it should really cite and compare to
the mixed membership stochastic blockmodel (MMSB):

Airoldi, E. M., Blei, D. M., Fienberg, S. E., & Xing, E. P. (2008). Mixed
membership stochastic blockmodels. Journal of Machine Learning Research.

The MMSB allows each node to randomly select a different "role" when deciding
whether to form each edge.

- General Discussion:

The aforementioned statistical models do not leverage text, and they do not use
scalable neural network implementations based on negative sampling, but they
are based on well-principled generative models instead of heuristic neural
network objective functions and algorithms.  There are more recent extensions
of these models and inference algorithms which are more scalable, and which do
leverage text.

Is the difference in performance between CENE and CANE in Figure 3
statistically insignificant? (A related question: were the experiments repeated
more than once with random train/test splits?)

Were the grid searches for hyperparameter values, mentioned in Section 5.3,
performed with evaluation on the test set (which would be problematic), or on a
validation set, or on the training set?